# Combination of Whole-Body Baseline CT Radiomics and Clinical Parameters to Predict Response and Survival in a Stage-IV Melanoma Cohort Undergoing Immunotherapy

**DOI:** 10.3390/cancers14122992

**Published:** 2022-06-17

**Authors:** Felix Peisen, Annika Hänsch, Alessa Hering, Andreas S. Brendlin, Saif Afat, Konstantin Nikolaou, Sergios Gatidis, Thomas Eigentler, Teresa Amaral, Jan H. Moltz, Ahmed E. Othman

**Affiliations:** 1Department of Diagnostic and Interventional Radiology, Eberhard Karls University, Tuebingen University Hospital, Hoppe-Seyler-Straße 3, 72076 Tuebingen, Germany; felix.peisen@med.uni-tuebingen.de (F.P.); andreas.brendlin@med.uni-tuebingen.de (A.S.B.); saif.afat@med.uni-tuebingen.de (S.A.); konstantin.nikolaou@med.uni-tuebingen.de (K.N.); sergios.gatidis@med.uni-tuebingen.de (S.G.); 2Fraunhofer Institute for Digital Medicine MEVIS, Max-von-Laue-Straße 2, 28359 Bremen, Germany; annika.haensch@mevis.fraunhofer.de (A.H.); alessa.hering@mevis.fraunhofer.de (A.H.); jan.moltz@mevis.fraunhofer.de (J.H.M.); 3Diagnostic Image Analysis Group, Radboudumc, Geert Grooteplein Zuid 10, 6525 GA Nijmegen, The Netherlands; 4Cluster of Excellence iFIT (EXC 2180) Image-Guided and Functionally Instructed Tumor Therapies (iFIT), University of Tuebingen, 72074 Tuebingen, Germany; 5Max Planck Institute for Intelligent Systems, Max-Planck-Ring 4, 72076 Tuebingen, Germany; 6Center of Dermato-Oncology, Department of Dermatology, Eberhard Karls University, Tuebingen University Hospital, Liebermeisterstraße 25, 72076 Tuebingen, Germany; thomas.eigentler@charite.de (T.E.); teresa.amaral@med.uni-tuebingen.de (T.A.); 7Department of Dermatology, Venereology and Allergology, Charité—Universitätsmedizin Berlin, Corporate Member of Freie Universität Berlin and Humbolt-Universität zu Berlin, Luisenstraße 2, 10117 Berlin, Germany; 8Institute of Neuroradiology, Johannes Gutenberg University Hospital Mainz, Langenbeckstraße 1, 55131 Mainz, Germany

**Keywords:** melanoma, prognostic biomarkers, imaging biomarkers, biomarkers for immunotherapy, checkpoint blockade, artificial intelligence and machine-learning

## Abstract

**Simple Summary:**

The use of immunotherapeutic agents significantly improved stage-IV melanoma patients’ overall progression-free survival. To identify patients who do not benefit from immunotherapy, both clinical parameters and experimental biomarkers such as radiomics are currently being evaluated. However, no radiomic biomarker is widely accepted for routine clinical use. In a large cohort of 262 stage-IV melanoma patients given first-line immunotherapy treatment, we investigated whether radiomics—based on the segmentation of all baseline metastases in the whole body—in combination with clinical parameters offered added value compared to the usage of clinical parameters alone in a machine-learning prediction model. The primary endpoints were response at three months, and survival rates at six and twelve months. The study indicated a potential, but non-significant, added value of radiomics for six-month and twelve-month survival prediction, thus underlining the relevance of clinical parameters.

**Abstract:**

Background: This study investigated whether a machine-learning-based combination of radiomics and clinical parameters was superior to the use of clinical parameters alone in predicting therapy response after three months, and overall survival after six and twelve months, in stage-IV malignant melanoma patients undergoing immunotherapy with PD-1 checkpoint inhibitors and CTLA-4 checkpoint inhibitors. Methods: A random forest model using clinical parameters (demographic variables and tumor markers = baseline model) was compared to a random forest model using clinical parameters and radiomics (extended model) via repeated 5-fold cross-validation. For this purpose, the baseline computed tomographies of 262 stage-IV malignant melanoma patients treated at a tertiary referral center were identified in the Central Malignant Melanoma Registry, and all visible metastases were three-dimensionally segmented (*n* = 6404). Results: The extended model was not significantly superior compared to the baseline model for survival prediction after six and twelve months (AUC (95% CI): 0.664 (0.598, 0.729) vs. 0.620 (0.545, 0.692) and AUC (95% CI): 0.600 (0.526, 0.667) vs. 0.588 (0.481, 0.629), respectively). The extended model was not significantly superior compared to the baseline model for response prediction after three months (AUC (95% CI): 0.641 (0.581, 0.700) vs. 0.656 (0.587, 0.719)). Conclusions: The study indicated a potential, but non-significant, added value of radiomics for six-month and twelve-month survival prediction of stage-IV melanoma patients undergoing immunotherapy.

## 1. Introduction

The treatment of patients with advanced melanoma has undergone a revolution in recent years with the introduction of new therapeutic approaches. On the one hand, blocking the RAF-RAS-MEK signaling pathway using combined BRAF and MEK inhibitors (so-called targeted therapy) is available; however, this is only possible for around 40% of patients due to the specific mutation patterns required [1,2,3,4,5]. On the other hand, since the introduction of the checkpoint inhibitors ipilimumab (CTLA-4) and nivolumab/pembrolizumab (PD-1) and their combination, effective immunotherapies have been available that enable treatment regardless of the mutation status [6]. The use of immunotherapeutic agents has significantly improved patients’ overall survival (OS) and progression-free survival (PFS) [7,8,9]. However, to date, around half of patients show primary resistance or develop secondary resistance during therapy. To identify patients who do not benefit from immunotherapy, only clinical parameters (such as lactate dehydrogenase (LDH)) and the presence of lung or liver metastases [10,11], as well as experimental biomarkers, (for example, those based on radiomics) are currently available. Radiomics aims to non-invasively extract phenotypic features from medical imaging by using automated algorithms, based either on manually programmed algorithms or deep learning, and subsequently attempts to develop imaging biomarkers from the derived features using machine- or deep-learning methods [12]. Radiomics has been used in some studies to generate added value for the prediction of OS and PFS [13,14,15,16,17,18]; however, no biomarker is widely accepted for routine clinical use [1].

Several issues repeatedly arise regarding radiological studies. First, the studies are often based on small cohorts and explicitly only use cohorts that are examined on a single device, with only one defined contrast-medium phase. Second, a follow-up examination is often required to document parameters such as tumor size reduction or the occurrence of new metastases. Third, most of the studies only segment the largest metastases according to RECIST 1.1. criteria or focus on a single organ, most likely due to the time-consuming process of manual segmentation. This has the drawback of a potential loss of information from the smaller metastases and parameters such as whole-body tumor burden.

Using a more extensive approach, we investigated whether radiomics, based on the segmentation of all baseline metastases in the whole body in combination with clinical parameters, offers added value compared to the usage of clinical parameters alone in a machine-learning prediction model. The primary endpoints were response at three months, and survival rates at six and twelve months.

## 2. Materials and Methods

### 2.1. Workflow Overview

The overall machine-learning (ML) workflow of this study is shown in Figure 1. For the selected patient cohort, all metastatic lesions were manually segmented in the baseline computed tomography (CT) images. Radiomic features were extracted and aggregated across the lesions per patient. The cohort was split into training and validation sets following a repeated 5-fold cross-validation scheme. An ML model, which consisted of feature pre-processing, feature selection, training, and validation, was applied and evaluated for clinical data only (baseline model) vs. clinical data and radiomic features (extended model).

### 2.2. Patient Selection

The Central Malignant Melanoma Registry (CMMR) was used to retrospectively identify patients diagnosed with stage-IV melanoma between 1 January 2015 and 31 December 2018. They were all given first-line treatment at the department of dermatology, a tertiary referral center for melanoma patients, with nivolumab or pembrolizumab mono (*n* = 146), or with a combination of nivolumab and ipilimumab (*n* = 116), according to current guidelines. The study protocol was approved by the institutional ethics board. Inclusion criteria were:Stage-IV melanoma;First-line treatment with a PD-1 checkpoint inhibitor, a CTLA-4 checkpoint inhibitor, or combination of both;Available baseline CT scans prior to treatment initiation;Available demographic data, follow-up data, and clinical metadata.

Exclusion criteria were an absence of CT baseline imaging, prior treatment with immunotherapy, and no visible metastasis on CT imaging. For a detailed illustration of the inclusion and exclusion process see Figure 2.

### 2.3. CT Imaging and Lesion Segmentation

Baseline CTs for all patients were identified in the local picture archiving and communication system (PACS), anonymized, and uploaded into a custom-made segmentation software (SATORI, Fraunhofer MEVIS, Bremen, Germany). In-house staging CTs were performed on four CT scanners (Sensation 64, SOMATOM Definition AS, SOMATOM Definition Flash, and SOMATOM Force—all from Siemens Healthineers, Erlangen, Germany) and one PET-CT scanner (Biograph128, from Siemens Healthineers, Erlangen, Germany). The in-house whole-body staging protocol was performed with a scan field from the skull base to the middle of the femur with patients laid in a supine position, arms raised above the head. Scanning was performed during the portal-venous phase after administration of body-weight-adapted contrast medium through the cubital vein. Attenuation-based tube-current modulation (CARE Dose, reference mAs 240) and tube voltage (120 kV) were applied. The following scan parameters were used:

SOMATOM Force: collimation, 128 × 0.6 mm; rotation time, 0.5 s; pitch 0.6.

Sensation64: collimation, 64 × 0.6 mm; rotation time, 0.5 s; pitch 0.6.

SOMATOM Definition Flash: collimation, 128 × 0.6 mm; rotation time, 0.5 s; pitch 1.0.

SOMATOM Definition AS: collimation, 64 × 0.6 mm; rotation time, 0.5 s; pitch 0.6.

Biograph128: collimation, 128 × 0.6 mm; rotation time, 0.5 s; pitch 0.8.

Slice thickness, as well as increment, were set to 3 mm. A medium-smooth kernel was used for image reconstruction.

Forty-three CTs of patients who had not received their baseline CT in-house at the department of diagnostic and interventional radiology, but at several external institutions, were also included to account for a more realistic sample and reduce sample bias. Detailed information for contrast-medium phase, tube current and tube voltage are not available for these cases. For a detailed distribution of CT vendors, see Appendix A.

The transverse reformatted images in the portal-venous phase were used for image analysis. All visible metastases were manually volumetrically segmented by a radiologist (FP with 4 years of experience in oncologic imaging) using SATORI (see Figure 3). In cases of uncertainty, a four-eyes principle with an experienced reader (AO with 8 years of experience in oncologic imaging) was applied.

### 2.4. Radiomic Feature Extraction and Aggregation

For each segmented lesion, 14 radiomic shape features, 18 first-order statistics features, and 75 texture features were extracted using the Pyradiomics software [12], which provides a reference implementation of the IBSI standard [19] with the documented deviations. The non-shape features were extracted as three different image types: original image (93 features), image filtered with Laplacian of Gaussian (LoG) with σ = 1, 2, 3, 4, and 5 mm (465 features), and wavelet-transformed image (744 features). In total, 1316 features were extracted per lesion. The features were not harmonized to account for different scanner types, as this has been shown to not improve model performance in previous work on CT-based features [20].

We aimed to perform patient-level training and outcome prediction. Each lesion-wise radiomic feature was aggregated per patient in four different ways: 1. feature value of the largest lesion; 2. volume-weighted mean of the feature value for the three largest lesions; 3. feature value of the most predictive lesion; and 4. volume-weighted mean of feature value for the three most predictive lesions. For determining the most predictive lesions, first, all lesions were ranked by their annotated type (from most to least predictive: liver, adrenal gland/heart/spleen, skeletal, lung, lymph nodes, and soft tissue/skin, based on the clinical experience of the T.A. and T.E.—both experts in the field of melanoma and experienced clinicians that have treated metastatic melanoma patients for many years at the Center of Dermato-Oncology, Tuebingen University Hospital—as well as on further evaluation of the data from Meier et al. [21]). Second, lesions were ranked per lesion type by their volume (larger lesions were rated as more predictive). In addition, the lesion count and total lesion volume were computed across all segmented lesions and per lesion type (liver, lung, lymph nodes, soft tissue/skin, skeletal, heart, spleen, adrenal gland, and other). In total, 5284 radiomic features were calculated per patient. Automatic feature selection was applied during training to select those features that had a high correlation with the outcome on training data, and little correlation with other selected features, using the fast correlation-based filter for feature selection (FCBF) method [22]. Compared to other methods such as minimum redundancy–maximum relevance (mRMR) feature selection [23], FCBF has the advantage of the number of selected features being chosen automatically. For FCBF, features were discretized by mapping the already normalized features to −1 for feature values below −0.5, to 0 for values between −0.5 and 0.5, and to 1 for higher feature values, as recommended for mRMR.

### 2.5. Machine-Learning Model

The machine-learning models were built for three different clinical endpoints: therapy response after three months according to RECIST 1.1 criteria (binarized: complete or partial response = response; stable or progressive disease = no response), and survival after six months and twelve months. Patients with PD after the first cycle of immunotherapy were considered non-responders. For each endpoint, the total patient cohort was reduced to those patients for whom the endpoint information was available. The excluded patients were censored.

Two ML models were trained per endpoint: the baseline model was trained on the clinical data, using all clinical features, as listed in Figure 3. The extended model was trained on all clinical features and a subset of the aggregated radiomic features, which was automatically selected per fold.

All ML models were trained in 10 × 5-fold cross-validation (CV) with random assignment of patients to the folds, to estimate the prediction performance. Per fold, the ML model pipeline consisted of four steps, of which steps 1–3 were performed based on the respective training set only:Pre-processing: Ordinal encoding of nominal clinical features [24], imputation of missing clinical feature values (0.5 for binary features, median for all other features), standard normalization (zero mean, unit variance) of all features;Feature selection using FCBF [22]: Applied only to radiomic features, clinical features were always used;Training: Fit of an extremely randomized forest [25];Validation: Prediction of outcome on the current validation set and comparison to true outcome using AUC.

Repeated CV was chosen over simple CV to reduce the impact of the random assignment to folds (or data-split) in the evaluation, and therefore, to obtain a more reliable estimate of the overall model performance.

A random forest (RF) was chosen as the core ML model because of its advantages such as the low need for hyper-parameter tuning and robustness to noisy variables [26]. We did not tune the model hyperparameters in a computationally expensive nested cross-validation loop, because the random forest is known to have low tunability [27] and to avoid overfitting of the hyperparameters to the comparably small training dataset.

A property of conventional random forests is the preferred splitting at variables with many possible values, such as continuous radiomic features, because more possible split points are available than for discrete or binary features. This bias in split-variable selection can lead to overfitting and splitting at non-optimal features within each tree. In the present study, most baseline clinical features had few discrete values, while the added radiomic features were multi-valued or continuous. Therefore, the extremely randomized forest variant [25] was chosen, which alleviates the bias in split selection by randomly binarizing all variables before applying the splitting rules. We used the implementation provided in the scikit-learn Phyton package (version 0.24.2, ExtraTreesClassifier) with default parameters, but enabled bootstrapping for building the trees [28].

### 2.6. Performance Evaluation

The area under the curve (AUC) of the receiver-operating-characteristic (ROC) curve was chosen as a classification performance metric. We used bootstrapping with 1000 samples to estimate a 95% confidence interval (CI) for the mean AUC of the 10 × 5-fold CV of each model. We computed the mean AUC by pooling the predictions on all 5 folds, and repeated this procedure for each of the 10 CV repetitions, using the same bootstrap sample on patient level per repetition [29]. Per bootstrap sample, we then calculated the mean AUC across the 10 repetitions. We computed a mean ROC curve analogously with 95% CI, by estimating the true positive rate of predictions via bootstrapping.

Statistically significant superior performance of the extended model is achieved if the CI of the mean AUC of the baseline and extended models do not overlap. Significant predictive capacity of a model following the outcome distribution is achieved if the lower bound of the CI is higher than 0.5.

Furthermore, we used the twelve-months survival prediction of the baseline model and the extended model to divide the cohort into low- vs. high-risk patients. A patient was classified as high-risk, when the twelve-months survival prediction of the model was negative, resulting in a risk stratification per model. The lifelines Python package (version 0.26.0) was employed to compute a Kaplan–Meier estimator for the overall survival of both groups, with a log-rank test for the statistical comparison of both groups [30].

## 3. Results

### 3.1. Demographics, Response after Three Months, and Survival after Six and Twelve Months

The patients were predominantly male (58%), with a median age of 70 years. The dominant histological subtypes were superficial spreading melanoma (27%) and nodular melanoma (24%). A total of 32% of the patients presented with elevated LDH and 48% with elevated S100 at baseline. For detailed demographic data, please refer to Table 1.

The median overall survival was 22.1 months (95% CI: 16.6–29.6), see Appendix A.

### 3.2. Prediction of Therapy Response and Survival Using Clinical Data and Radiomic Features

#### 3.2.1. Machine-Learning Model Comparison

The comparison of the two different models (baseline model using only clinical data and extended model using clinical data and radiomic parameters) showed that the extended model was numerically superior to the baseline model for survival prediction, but not for response prediction (delta of the AUC for response after three months, −0.015; delta for survival after six months, +0.044; delta for survival after twelve months, +0.042), see Table 2. However, the superiority was not significant, as there was an overlap of the confidence interval for all endpoints. Figure 4 shows the mean ROC curve for all endpoints and both models, where an overlap of the CI of the ROC curves can also be seen.

Except for the baseline model for survival after twelve months, all models indicated a significant prediction capacity (AUC > 0.5) (see Table 2).

When the mean AUC and CI were computed for a single 5-fold CV instead of the repeated 10 × 5-fold CV, the mean AUC was improved by the extended model in 3 out of 10 repetitions for response prediction, 10 out of 10 repetitions for survival after six months, and 8 out of 10 repetitions for survival after twelve months, depending on the random data-split of each CV.

#### 3.2.2. Feature Selection

In a 10 × 5-fold cross-validation of 5284 radiomic features, on average, 6.6 features were chosen for response after three months, 7.4 features for survival after six months, and 6.8 features for survival after twelve months.

### 3.3. Low-Risk vs. High-Risk Stratification for Twelve-Month Survival Prediction

The subset of the patient cohort for which twelve-month survival data was available was split into a low-risk and a high-risk group with Kaplan–Meier estimators, depending on twelve-month survival prediction in the 5-fold cross-validation (repeat 0). Log-rank tests revealed a borderline significant distinction for the baseline model (*p* = 0.08) as well as the extended model (*p* = 0.06). Please see Figure 5 for the comparison.

## 4. Discussion

### 4.1. Prediction of Therapy Response

Based on a 10 × 5-fold CV, the baseline model and extended model indicated a significant predictive capacity (AUC > 0.5). The baseline model was numerically superior to the extended model (AUC 0.656 vs. 0.641), although without a significant difference (CI overlap). The results revealed that radiomic parameters from pre-treatment baseline CTs did not add significantly more information to the prediction of response after three months, compared to a prediction model based on baseline clinical parameters alone. We even found that adding non-predictive features can potentially deteriorate the performance of the model.

### 4.2. Prediction of Six-Month and Twelve-Month Survival

Like for therapy response, a baseline model (clinical features) was compared to an extended model (clinical and radiomic features) for the binary endpoints of six-month and twelve-month survival. For six-month survival, the baseline and extended model indicated a significant predictive capacity (AUC > 0.5). For twelve-month survival, the baseline model did not indicate a significant predictive capacity (AUC 0.5 included in CI), but the extended model did. The extended model showed numerical superiority (AUC 0.664 vs. 0.620 for six-month survival and AUC 0.600 vs. 0.558 for twelve-month survival); however, the extended model did not reach a significant difference for either endpoint (CI overlap).

For response prediction, as well as for survival prediction, both models presented a large range of mean AUC on a single 5-fold CV out of 10 repetitions, depending on the random data-split. With a single split into training, validation, and test set, which gives only one data point as opposed to a distribution of possible AUC values, this observation would not have been possible. Therefore, we advocate for the use of (repeated) cross-validation instead of a single data-split for the evaluation of machine-learning models [31].

The additional amount of information gained by the extended prediction model is limited, as the low AUCs and delta AUCs of the results indicate. CT parameters such as voltage, tube current, pitch, contrast-medium phase, reconstruction kernel, and slice thickness vary between institutions and scanners, with consequent impacts on the radiomic parameters. Some radiomic studies do not pool patients from different locations for the training dataset, but carefully select a cohort from one or two defined tomographs with predefined scan parameters. This, however, does not reflect the reality of tumor boards, where image data from different sources is discussed, especially at baseline diagnostics, when patients have been diagnosed externally and no in-house image data are available [32,33,34]. Another explanation for the low AUCs might be that information about the therapy response per lesion was missing, as the study aimed to use baseline CTs only.

### 4.3. Risk Stratification for Twelve-Month Survival

Both models reached borderline significance in differentiating a low-risk cohort from a high-risk cohort for twelve-month survival, with the extended model performing with slight superiority over the baseline model (*p* = 0.06 vs. *p* = 0.08). This is in line with a publication by Durot et al., who showed, in a small cohort with a more basic statistical analysis, that texture parameters were significantly associated with both lower OS and lower PFS after the administration of pembrolizumab [13].

### 4.4. Feature Selection/Radiomic Biomarker

In a 10 × 5-fold CV of more than five thousand radiomic features, on average, 6.6 features were chosen for response after three months, 7.4 features for survival after six months, and 6.8 features for survival after twelve months. The features chosen were dependent on the current data-split, thus meaning that there was no stable radiomic biomarker fitting all cross-validation splits. This effect has previously been described in the literature. Radiomic features are known to be highly correlated, meaning that certain features can be used interchangeably without affecting predictive performance and, therefore, different features may be randomly selected in different splits [35]. Therefore, it is not productive to describe differences in selected features in a 10 × 5-fold CV and to report single chosen features. Permutation feature importance—a model-inspection technique for machine-learning models, such as random forests, that investigates the relationship between a feature and a target and shows how much the model depends on the feature [26]—was not possible, because the technique requires uncorrelated features. As we only performed feature selection on radiomic parameters, the clinical parameters may have been correlated; therefore, permutation feature importance would not have been meaningful. Lastly, as discussed before, the AUCs were not very high (<0.7), which serves as an indicator that the selected radiomic features cannot serve as a stable biomarker.

A potential explanation might be the initial heterogeneity of the study cohort or the method of patient-wise feature aggregation. Some radiomic studies do not pool patients from different locations for the training dataset, but carefully select a cohort from one or two defined tomographs with predefined scan parameters. This, however, does not reflect the reality of tumor boards, where image data from different sources are discussed, especially at baseline diagnostics, when patients have been diagnosed externally and no in-house image data are available.

### 4.5. Strengths

The study used a large cohort (262 patients, 6404 segmented metastases) with prospective documentation of clinical data and long-term-follow ups. The cohort consisted of a first-line-treatment collective and showed a typical response pattern after three months of immune therapy (CR 4%, PR 27%, SD 16%, and PD 37%) [1]. Six-month survival and twelve-month survival, as well as median overall survival, were also in the typical range of a stage-IV melanoma cohort (69%, 44% and 22 months, respectively) [1,36].

We aimed to test the applicability of a random forest model in a real-life cohort, including several CT vendors and different sources of image data to reduce selection bias. More importantly, the study followed a whole-body segmentation approach and used all visible metastases of the baseline CTs. In some studies, radiomics has been proven to generate additional information for the prediction of response and overall survival of stage-IV melanoma patients undergoing immune therapy [14,15,16,17,18,37]. Unfortunately, many of those studies use very homogenous patient cohorts, lacking proof of usability in a real-life scenario [16]. Although the clinical application of a potential biomarker would be challenging with a whole-body segmentation approach, the study’s aim was to use as much information about the metastatic load as possible to create a benchmark for predictive performance. Furthermore, due to modern (semi)automated segmentation tools, whole-body segmentation will become faster and easier to apply [38,39,40].

### 4.6. Limitations

The present collective represented a very large cohort; nevertheless, an extended number of patients would have been of value to the study. Unfortunately, stage-IV melanoma patients with complete documented clinical information and simultaneously available image data are hard to recruit, and the initial full-body-segmentation approach did not allow for the further inclusion of more patients.

Secondly, the study lacks an external validation cohort. We solved this problem through multiple cross-validations, an established method that has been used in prior studies [41]. In our case, correction for multiple testing was waived because most of the results were not statistically significant.

Thirdly, the study included 43 CTs from external sources and twenty different tomographs from five vendors, partially lacking detailed information of CT acquisition parameters. This drawback hinders a more detailed evaluation of the influence of varying acquisition parameters on radiomics; however, that constraint was accepted to reduce potential selection bias and guarantee a realistic real-life sample as requested by radiological guidelines [42].

Fourthly, the lesion segmentation was manually carried out only once by F.P. under supervision of A.O., lacking a second reading and a second reader; this was due to the immense number of metastases following a whole-body segmentation approach. An intraclass correlation coefficient analysis was, therefore, not possible, and an evaluation of inter- and intra-reader variability is missing.

## 5. Conclusions

The present study indicated the potential, albeit non-significant, added value of radiomics for six-month and twelve-month survival prediction in stage-IV melanoma patients undergoing immunotherapy. While the study followed a whole-body segmentation approach of all visible metastases to gain as much information as possible, the resulting AUCs remained low. Further approaches integrating other biomarkers or radiomic techniques are needed to improve the prediction ability of the current model and others.

## Figures and Tables

**Figure 1 cancers-14-02992-f001:**
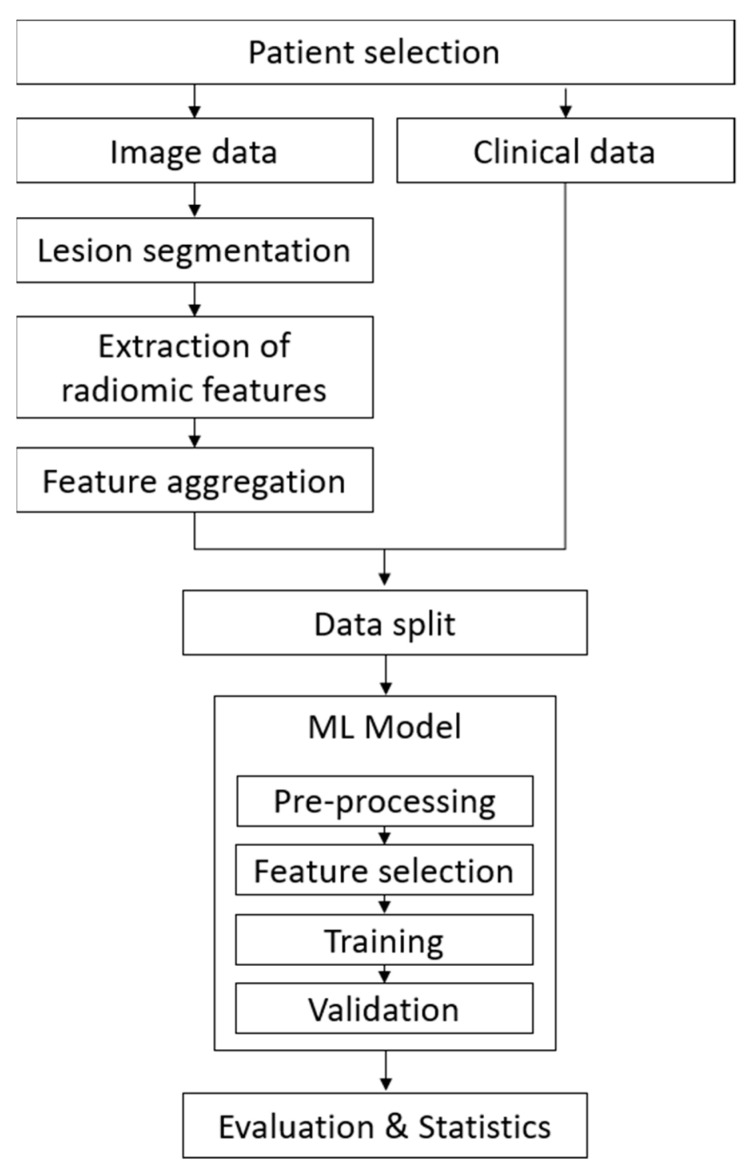
Schematic overview of the data processing and machine-learning workflow.

**Figure 2 cancers-14-02992-f002:**
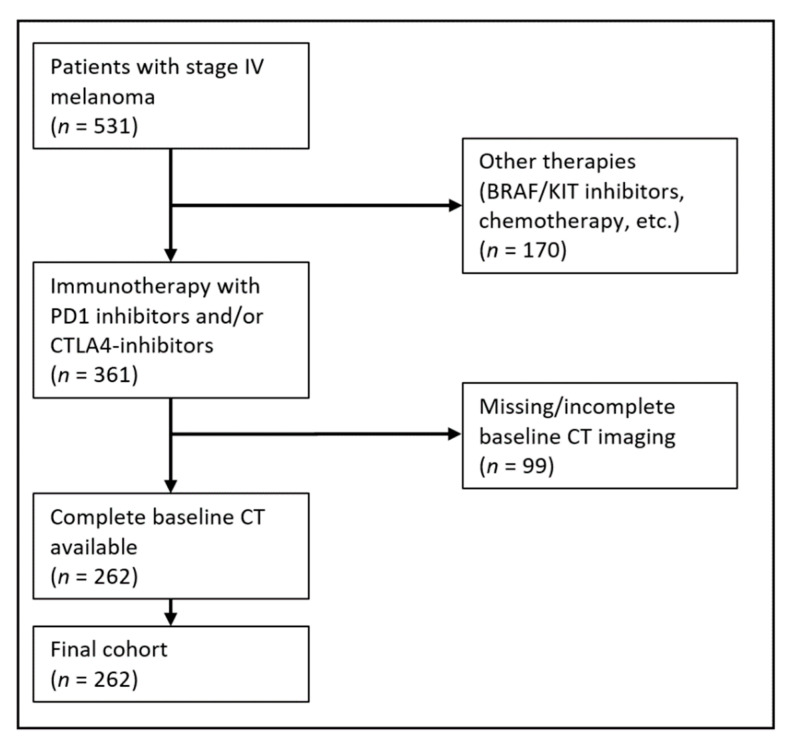
Patient selection.

**Figure 3 cancers-14-02992-f003:**
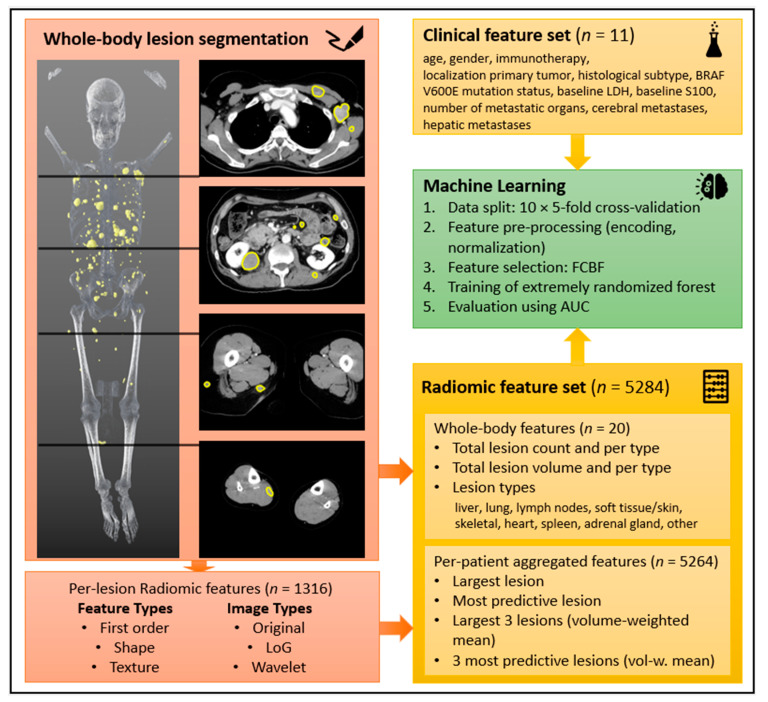
Schematic feature extraction and machine-learning workflow: Top left—3D reformatted model of all segmented metastases (yellow) and examples of 2D segmentation process in axial reformatted CT slices in portal-venous contrast-medium phase; bottom left—radiomic feature types; top right—clinical feature set; bottom right—radiomic feature set; middle right—overview of machine-learning process. Abbreviations: AUC—area under the curve; BRAF—v-Raf murine sarcoma viral oncogene homolog B1; FCBF—fast correlation based filter; LDH—lactate dehydrogenase; LoG—Laplacian of Gaussian.

**Figure 4 cancers-14-02992-f004:**
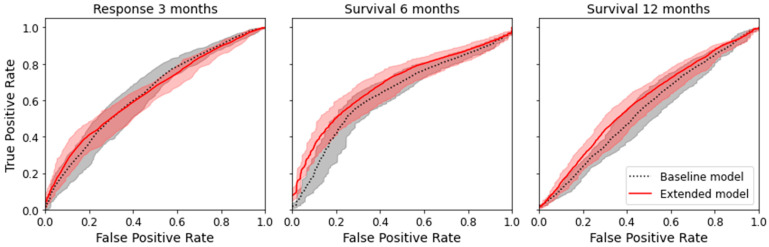
Mean ROC curves with 95% confidence interval for the true positive rate computed by bootstrapping the 10 × 5-fold CV.

**Figure 5 cancers-14-02992-f005:**
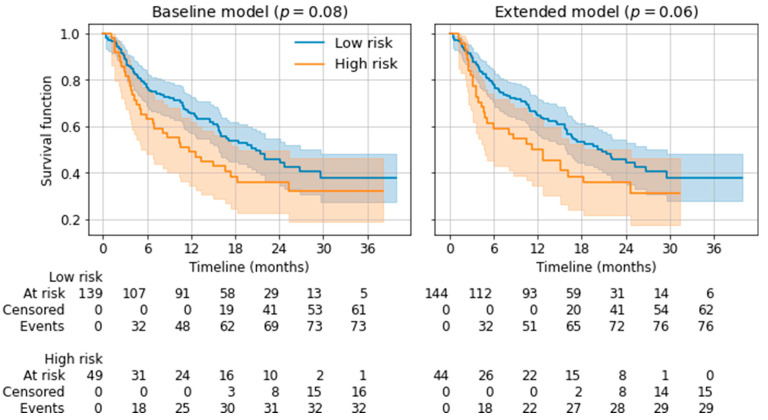
Kaplan–Meier estimators for low-risk and high-risk groups based on the predicted 12-month survival. *P*-values from log-rank tests are given for the distinction of both risk groups.

**Table 1 cancers-14-02992-t001:** Patients’ characteristics.

**Clinical Data**		
Age (years) [median, (IQR)]		70 (22)
Gender (female) [*n*, %]		109 (42%)
Localization of primary tumor [*n*, %]	Head/neck	50 (19%)
	Torso	63 (24%)
	Upper extremity	30 (11%)
	Lower extremity	71 (27%)
	Other	13 (5%)
	n/a	35 (13%)
Histological subtype [*n*, %]	SSM	71 (27%)
	NM	62 (24%)
	LMM	13 (5%)
	ALM	29 (11%)
	Mucosal	13 (5%)
	Occult	61 (23%)
	n/a	13 (5%)
BRAF V600E mutation status [*n*, %]	BRAF wildtype	180 (69%)
	BRAF mutation	74 (28%)
	n/a	8 (3%)
Baseline LDH [*n*, %]	Normal (<250 U/l)	164 (63%)
	Elevated (≥250 U/l)	85 (32%)
	n/a	13 (5%)
Baseline S100 [*n*, %]	Normal (<0.1 µg/l)	117 (45%)
	Elevated (≥0.1 µg/l)	125 (48%)
	n/a	20 (8%)
Number of metastatic organs [*n*, %]	1–3	232 (89%)
	>3	30 (11%)
Cerebral metastases [*n*, %]		48 (18%)
Hepatic metastases [*n*, %]		85 (32%)
Immunotherapy [*n*, %]	PD1	146 (56%)
	PD1+CTLA4	116 (44%)
**Patient Outcome**		
Response after 3 months (RECIST 1.1) [*n*, %]	CR	10 (4%)
	PR	72 (27%)
	SD	42 (16%)
	PD	96 (37%)
	n/a	42 (16%)
Survival after 6 months [*n*, %]	Yes	181 (69%)
	No	49 (19%)
	n/a	32 (12%)
Survival after 12 months [*n*, %]	Yes	115 (44%)
	No	73 (28%)
	n/a	74 (28%)
Lesion counts [*n* lesions, *n* patients]	All	6404, 262
	Lung	2738, 157
	Liver	1120, 79
	Soft tissue/skin	1111, 110
	Lymph nodes	876, 154
	Skeletal	172, 42
	Spleen	97, 12
	Heart	8, 3
	Other	238, 54

Abbreviations: ALM—acral lentiginous melanoma; CR—complete response; CTLA4—cytotoxic T-lymphocyte-associated protein 4; IQR—interquartile range; LDH—lactate dehydrogenase; LMM—lentigo maligna melanoma; n/a—not available; NM—nodular melanoma; PD1—programmed death 1; PD—progressive disease; PR—partial response; RECIST—Response Evaluation Criteria in Solid Tumors; SD—stable disease; SSM—superficial spreading melanoma.

**Table 2 cancers-14-02992-t002:** Number of cases with class distributions, and mean AUC from a 10 × 5-fold CV and 95% confidence interval computed by bootstrapping the 10 × 5-fold CV.

	Binary Endpoint		
	Response at 3 Months	Survival at 6 Months	Survival at 12 Months
*n* cases, (class 0, class 1)	220 (138, 82)	230 (49, 181)	188 (73, 115)
Baseline model (clinical features), (AUC (95% CI))	0.656 (0.587, 0.719)	0.620 (0.545, 0.692)	0.558 (0.481, 0.629)
Extended model (clinical and radiomic features), (AUC (95% CI))	0.641 (0.581, 0.700)	0.664 (0.598, 0.729)	0.600 (0.526, 0.667)

Abbreviations: AUC—area under the curve; CI—confidence interval; CV—cross-validation; *n*—number.

## Data Availability

Data may be made available after a reasonable and well-justified request to Ahmed E Othman. Data cannot, however, be made freely available to the public, due to privacy regulations. The codes and materials used in this study may be made available for the purposes of reproducing or extending the analysis, pending material-transfer agreements.

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
