# Peer review of "Combination of Whole-Body Baseline CT Radiomics and Clinical Parameters to Predict Response and Survival in a Stage-IV Melanoma Cohort Undergoing Immunotherapy"

_cancers, 2022, doi:10.3390/cancers14122992_

Round 1
Reviewer 1 Report
In this study, the authors evaluate the utility of combining radiomics and clinical parameters in predicting response to first line immune checkpoint inhibition in patients with stage IV melanoma.
Despite this study being negative, the idea is novel and interesting. The results would add value to the existing literature.
Tables and figures are particularly clear. Limitations are well-defined.
I would suggest the following edits:
-"Furthermore, we used the twelve months’ survival prediction of the baseline model and the extended model to divide the cohort into low- vs. high-risk patients according to each model.": I would suggest elaborating further on how patients were categorized high vs low risk (statistically and clinically).
-The KM curves of low vs high risk should be added to the main manuscript and not as supplementary files
-It would be adequate to clearly mention in the methods section that RECIST 1.1 criteria were used.
-Were patients with PD after the first cycle considered non responders? or were they re-evaluated after the second cycle? This determination should be clear.
Author Response
Response to comments of Reviewer #1:
Point 1: "Furthermore, we used the twelve months’ survival prediction of the baseline model and the extended model to divide the cohort into low- vs. high-risk patients according to each model .": I would suggest elaborating further on how patients were categorized high vs low risk (statistically and clinically).
- We would like to thank the reviewer for that remark. We only categorized the patients based on the model prediction, not from a clinical perspective. We added a sentence in section 2.6, hopefully making our approach clearer: “A patient was classified as high-risk, when the twelve-months’ survival prediction of the model was negative, resulting in a risk stratification per model”.
Point 2: The KM curves of low vs high risk should be added to the main manuscript and not as supplementary files.
- Thank you very much for that suggestion. The Kaplan-Meier estimators for low-risk and high-risk groups are now included in the main manuscript in section 3.3. as figure 5.
Point 3: It would be adequate to clearly mention in the methods section that RECIST 1.1 criteria were used.
- We would like to thank the reviewer for this important point. In section 2.5. we now clearly indicate that RECIST 1.1. criteria were used for response assessment.
Point 4: Were patients with PD after the first cycle considered non responders? or were they re-evaluated after the second cycle? This determination should be clear.
- We apologize that this information was not initially included. In section 2.5. we now clarify that patients with PD after the first cycle of immunotherapy were considered non responders, according to RECIST 1.1.
We hope that the applied changes regain your agreement. In case that further clarifications or information should be needed, please do not hesitate to contact us any time.
Yours sincerely
Felix Peisen
Reviewer 2 Report
In this work the Authors aim at evaluating radiomic features as novel biomarkers to improve the ability of the current models in predicting therapy response, overall and free survival in stage IV melanoma patients. The authors are correct in stressing the importance of the clinical need under investigation, but some changes should be addressed.
Major revisions
In page 4, the sentence: “Forty-three external baseline CTs were also included to account for a more realistic sample and reduce sample bias. Detailed information for contrast medium phase, tube current and tube voltage are not available for these cases.” Could you please provide more details about this approach? Please specify what you mean with “external”? Which patients were scanned?
Since you use such a high number of different tomographs from different vendors, why didn’t you perform any kind of image harmonization?
You write “features were extracted using the Pyradiomics software [12], which currently serves as the reference standard for radiomics analysis” (page 5). However, the reference standard for radiomics is the IBSI, and not all features extracted from Pyradiomics are IBSI-compliant. Therefore, I would modify the sentence and specify if all the extracted features in the study were or were not IBSI-compliant.
Page 6: did you discretize data before applying the Fast Correlation Based Filter for Feature Selection (FCBF) method? How?
When imputing missing data, don’t you think that the 0.5 value for binary features could have artificially hampered the analysis? Did you tried to perform the same analysis also without missing data?
Page 6: why didn’t you tried to optimize random forest hyperparameters
How many features were used as input for each prediction? As 11 clinical features + 6/8 radiomic features is a quite high number of inputs for a ML model, especially compared to the number of data available… don’t you think that this aspect could have affected the models’ ability?
Minor revisions
For the segmentation step, did you use a size cut-off based on the image resolution or did you segment all the smallest metastases too? How do you justify this choice?
Page 5: When you write “For determining the most predictive lesions, all lesions were first ranked by their annotated type (from most to least predictive: liver, ad- renal gland/heart/spleen, skeletal, lung, lymph nodes, soft tissue/skin)”, can you better specify the choice of this ranking criterion
Page 9, 3.2.1. section: there is a typo “Error! Reference source not found.” in the text
Page 12, 4.6 Limitations: I would specify the lack of ICC analysis to evaluate the inter- and intra-observer variability in segmenting the lesions
Author Response
Response to comments of Reviewer #2:
Major revisions
Point 1: In page 4, the sentence: “Forty-three external baseline CTs were also included to account for a more realistic sample and reduce sample bias. Detailed information for contrast medium phase, tube current and tube voltage are not available for these cases.” Could you please provide more details about this approach? Please specify what you mean with “external”? Which patients were scanned?
- We thank you for that question. This is a point that need clarification in the manuscript. Some patients undergoing first-line immunotherapy at the Department of Dermatology University Hospital Tübingen received their pre-treatment baseline CT not inhouse in the Department for Diagnostic and Interventional Radiology but in several external institutions and were then consecutively assigned to the Department of Dermatology University Hospital Tübingen for treatment. Despite the missing detailed information for contrast medium phase, tube current and tube voltage we decided to include externally acquired baseline CTs of those patients in our analysis, as they received the same therapy regimens as patients that underwent inhouse baseline CT imaging. Follow up imaging was then carried out inhouse at the Department of Diagnostic and Interventional Radiology University Hospital Tübingen. Our rationale was to cover a real-life scenario and reduce sample bias, as postulated by the RSNA (“…Use multivendor images, preferably for each phase of the AI evaluation (training, validation, test sets). …scans from one vendor do not look like those from another vendor. Such differences are detected by radiomics and AI algorithms. Vendor-specific algorithms are of much less interest than multivendor AI algorithms.” Bluemke DA, Moy L, Bredella MA, Ertl-Wagner BB, Fowler KJ, Goh VJ, Halpern EF, Hess CP, Schiebler ML, Weiss CR. Assessing Radiology Research on Artificial Intelligence: A Brief Guide for Authors, Reviewers, and Readers-From the Radiology Editorial Board. Radiology. 2020 Mar;294(3):487-489. doi: 10.1148/radiol.2019192515. Epub 2019 Dec 31. PMID: 31891322.).
We have reformatted the sentence in section 2.3. and hope that the information is now presented in a more concise way: “Forty-three CTs of patients having received their baseline CT not inhouse at the department of diagnostic and interventional radiology, but at several external institutions, were also included to account for a more realistic sample and reduce sample bias.”.
Point 2: Since you use such a high number of different tomographs from different vendors, why didn’t you perform any kind of image harmonization?
- Thank you for this interesting question. In our experience and as published in previous work by parts of our team, feature harmonization does not improve performance for CT-based features (please compare Enke, J.S.; Moltz, J.H.; D'Anastasi, M.; Kunz, W.G.; Schmidt, C.; Maurus, S.; Mühlberg, A.; Katzmann, A.; Sühling, M.; Hahn, H.; Nörenberg, D.; Huber, T. Radiomics Features of the Spleen as Surrogates for CT-Based Lymphoma Diagnosis and Subtype Differentiation. Cancers 2022, 14, 713. https://doi.org/10.3390/cancers14030713). We added a respective sentence in section 2.4. and provide a reference to justify our chosen approach: “The features were not harmonized to account for different scanner types, as this has shown to not improve the model performance in previous work for CT-based features.”
Point 3: You write “features were extracted using the Pyradiomics software [12], which currently serves as the reference standard for radiomics analysis” (page 5). However, the reference standard for radiomics is the IBSI, and not all features extracted from Pyradiomics are IBSI-compliant. Therefore, I would modify the sentence and specify if all the extracted features in the study were or were not IBSI-compliant.
- We would like to thank the reviewer very much for this point. We clarified this by reformulating the sentence in section 2.4. and provide corresponding references: “For each segmented lesion, 14 radiomic shape features, 18 first order statistics features, and 75 texture features were extracted using the Pyradiomics software, which provides a reference implementation of the IBSI standard with the documented deviation.”
Point 4: Page 6: did you discretize data before applying the Fast Correlation Based Filter for Feature Selection (FCBF) method? How?
- Thank you very much for pointing out that missing information. Indeed, for FCBF features were discretized. We added this information at the end of section 2.4.:” For FCBF, features were discretized by mapping the already normalized features to -1 for feature values below -0.5, to 0 for values between -0.5 and 0.5, and to 1 for higher feature values, as recommended for mRMR.”
Point 5: When imputing missing data, don’t you think that the 0.5 value for binary features could have artificially hampered the analysis? Did you tried to perform the same analysis also without missing data ?
- We thank the reviewer for that very interesting question and would like to clarify our approach. For binary values, we choose to impute using the constant 0.5 instead of the median, so that the random forest is able decide how to split the data and not always assign the cases to the majority class for that variable. We did not try to perform the analysis without missing data, because the initial dataset size was already limited with 262 cases. In total, only three variables are affected by this imputation method (the other binary ones are complete): BRAF mutation status (8 missing values), elevated LDH (13 missing values) and elevated S100 (20 missing values), so that we expect the impact of a different imputation to be limited.
Point 6: Page 6: why didn’t you tried to optimize random forest hyperparameters
- We agree with the reviewer that this point needs further clarification. In our experience a random forest is known to have a low tunability. Furthermore, to avoid overfitting of the hyperparameters to the comparably small training data set we did not perform tuning. We added a sentence to clarify this in section 2.5. and provide relevant literature: “We did not tune the model hyperparameters in a computationally expensive nested cross-validation loop because the random forest is known to have a low tunability and to avoid overfitting of the hyperparameters to the comparably small training data set.”
Point 7: How many features were used as input for each prediction? As 11 clinical features + 6/8 radiomic features is a quite high number of inputs for a ML model, especially compared to the number of data available… don’t you think that this aspect could have affected the models’ ability?
- In preliminary experiments, applying FCBF to the clinical features only in the baseline model (without Radiomics) led to worse performance. Therefore, we decided to always use all clinical features and extend the model with the automatically selected Radiomics features. We also chose Random Forest because it is known to work well with larger number of variables compared to other ML methods (please compare Biau, G., Scornet, E. A random forest guided tour. TEST 25, 197–227 (2016). https://doi.org/10.1007/s11749-016-0481-7).
Minor revisions
Point 8: For the segmentation step, did you use a size cut-off based on the image resolution or did you segment all the smallest metastases too? How do you justify this choice?
- We used all segmented metastases for the analysis. We are aware that computation of radiomic features does not make sense when lesions are too small. However, in the feature aggregation step we use only the up to three largest or most predictive (also ranked by size) lesions. The smaller lesions only add to the overall lesion count which in our view does not comprise feature calculation.
Point 9: Page 5: When you write “For determining the most predictive lesions, all lesions were first ranked by their annotated type (from most to least predictive: liver, ad- renal gland/heart/spleen, skeletal, lung, lymph nodes, soft tissue/skin)”, can you better specify the choice of this ranking criterion
- The ranking of the prognosis of the various metastasis localizations was based on clinical experience by Prof. Eigentler and Dr. Amaral, both experts in the field of melanoma and experienced clinicians, that have treated metastatic melanoma patients since many years at the Center of Dermato-Oncology, Department of Dermatology, Eberhard Karls University, Tuebingen University Hospital as well as on further evaluation of the data from Meier et al. (Meier, F., Will, S., Ellwanger, U., Schlagenhauff, B., Schittek, B., Rassner, G. and Garbe, C. (2002), Metastatic pathways and time courses in the orderly progression of cutaneous melanoma. British Journal of Dermatology, 147: 62-70. https://doi.org/10.1046/j.1365-2133.2002.04867.x). In a series of 3001 patients, they performed a detailed analysis of the different metastatic pathways, the time course of the development of metastases and the factors influencing them. This information is now specified in section 2.4.: “…from most to least predictive: liver, adrenal gland/heart/spleen, skeletal, lung, lymph nodes, soft tissue/skin, based on clinical experience by T.A. and T.E., both experts in the field of melanoma and experienced clinicians, that have treated metastatic melanoma patients since many years at the Center of Dermato-Oncology Tuebingen University Hospital as well as on further evaluation of the data from Meier et al…”.
Point 10: Page 9, 3.2.1. section: there is a typo “Error! Reference source not found.” in the text
- Thank you very much for this information. The typo was corrected and now refers to figure 4.
Point 11: Page 12, 4.6 Limitations: I would specify the lack of ICC analysis to evaluate the inter- and intra-observer variability in segmenting the lesions
- We would like to thank the reviewer for this suggestion. We have included the lack of ICC analysis in the limitations section and now clearly state a missing evaluation of inter- and intra-observer variability: “Fourthly, the lesion segmentation was manually carried out only once by F.P. under supervision of A.O., lacking a second reading and a second reader, due to the immense number of metastases following a whole-body segmentation approach. An intraclass correlation coefficient analysis was therefore not possible, and evaluation of inter- and intra-reader variability is missing.”
We hope that the applied changes regain your agreement. In case that further clarifications or information should be needed, please do not hesitate to contact us any time.
Yours sincerely
Felix Peisen
Round 2
Reviewer 2 Report
Thank you for the replies.